# Facile Preparation of Cellulose Aerogels with Controllable Pore Structure

**DOI:** 10.3390/nano13030613

**Published:** 2023-02-03

**Authors:** Jiahao Qiu, Xingzhong Guo, Wei Lei, Ronghua Ding, Yun Zhang, Hui Yang

**Affiliations:** 1State Key Laboratory of Silicon Materials, School of Materials Science and Engineering, Zhejiang University, Hangzhou 310027, China; 2Hangzhou Global Scientific and Technological Innovation Center, Zhejiang University, Hangzhou 311200, China; 3Pan Asia Microvent Tech (Jiangsu) Corporation & Zhejiang University Micro-Nano-Porous Materials United Research Development Center, Changzhou 213100, China

**Keywords:** cellulose aerogels, sol-gel method, temperature control, micromorphology

## Abstract

Cellulose aerogels are the latest generation of aerogels and have also received extensive attention due to their renewable and biocompatible properties. Herein, cellulose aerogel was facilely prepared by using NaOH/urea solution as solvent, raising the temperature to control gelation and drying wet gel sequentially. With NaOH/urea solution as solvent, the cellulose concentration has an important impact on the micromorphology of cellulose aerogels, while the aging time rarely affects the micromorphology. The appropriate solvent and drying method allow the formation of different cellulose crystalline structures. Different from the Cellulose Ⅰ crystalline structure of raw cellulose powder, the cellulose phase of as-prepared cellulose aerogels belongs to the Cellulose Ⅱ crystalline structure, and to some extent the pyrolysis temperature is also lower than that of raw cellulose powder. The resultant cellulose aerogel prepared by using NaOH/urea solution as solvent and freeze-drying has a uniform macroporous structure with a macropore size of 1~3 µm.

## 1. Introduction

First-generation aerogels are known as silica aerogel, and they were synthesized by sol-gel method and drying under supercritical conditions in 1931 [1]; furthermore, silica aerogels have been widely studied and applied because of their light weight (as low as 0.1 g·cm^−1^), high specific surface area (800~1000 m^2^·g^−1^) and low thermal conductivity (as low as 0.017 W·m^−1^·k^−1^) [2,3,4,5,6]. Second-generation aerogels are metal oxide (zirconia [7], alumina [8], titanium oxide [9], etc.) aerogels and have been extensively researched due to their high temperature performance and superior functional features. Cellulose aerogels are considered to be the latest generation of aerogels [10]. The cellulose aerogel is renewable, biodegradable, with a high porosity, high specific surface area, low density and 3D interconnected porous network structures [11,12,13], and it can be widely used in adsorption, separation, thermal insulation, biomedicine and other fields [14,15,16,17].

The problem of energy exhaustion and environmental pollution has become increasingly prominent and has promoted research on biodegradable, cheap and non-toxic natural polymer materials [18]. The dissolution of cellulose has always been one of the difficulties in the preparation of cellulose aerogels. In 1850, cellulose was dissolved for the first time. Since then, N,N-dimethylacetamide (DMAc)/LiCl [19], N,N-dimethylformamide (DMF)/N_2_O_4_ [20], dimethyl sulfoxide (DMSO) [21], etc. have been found to dissolve cellulose well, but these solvents will cause environmental pollution. Ionic liquids have been used for a long time as cellulose solvents to prepare cellulose aerogels [22]. In an ionic liquid system, there exists no gelation process, and the network structure of cellulose aerogels is formed through non-solvent induced phase separation [11]. However, the high cost of ionic liquid has limited the industrialization of cellulose aerogels. NaOH/urea solution can rapidly dissolve cellulose at a low temperature [23] and replace the ionic liquid owing to its low cost and no pollution. In current research, it is most widely used as a solvent to dissolve cellulose. At present, the cellulose sol formed in NaOH/urea solution can be gelated by adding a cross-linking agent [24], adjusting pH, etc. Most studies will choose to add a cross-linking agent to obtain cellulose gel, while there are few systematic studies on gelation by temperature control.

In this study, we demonstrate the facile preparation of cellulose aerogel with a controllable pore structure. The NaOH/urea solution is used as an environmentally friendly and low-cost solvent to dissolve raw cellulose powder to form the cellulose sol, and the gelation of the formed cellulose sol can be controlled by raising the temperature. Freeze-drying or supercritical drying is used to dry the wet gel to prepare regenerated cellulose aerogel. The micromorphology and pore-structure characteristics of cellulose aerogels prepared under different conditions are investigated in detail and compared with each other. This environmentally friendly and low-cost preparation approach shows promise for applications in the large-scale preparation of cellulose aerogels.

## 2. Materials and Methods

### 2.1. Materials

Cellulose powder was purchased from Shanghai Aladdin Biochemical Technology Co., Ltd. (Shanghai, China). 1-Ethyl-3-methylimidazolium Chloride ([AMIm]Cl) was purchased from Beijing Huawei Ruike Chemical Co., Ltd. (Beijing, China). Sodium hydroxid (NaOH) and Urea were purchased from Sinopharm Chemical Reagent Co., Ltd. (Shanghai, China). Anhydrous ethanol was purchased from Sinopharm Chemical Reagent Co., Ltd. (Shanghai, China). The reagents and solvents in the study were analytical reagents (AR) without further purification.

### 2.2. Preparation

The preparation of cellulose aerogels with NaOH/urea solution as solvent was described as follows. A total of 18 mL deionized water, 1.2 g sodium hydroxide and 0.8 g urea were mixed in the beaker to prepare 6 wt% NaOH/4 wt% urea solution. Then, the cellulose powder was added into the solution, and the mixture was stirred at −5 °C for 60 min to form sol. The sol was put in the oven for gelation and aging by raising the temperature. After solvent exchange with water 3 times at ambient temperature, the cellulose hydrogel was quickly frozen by liquid nitrogen and freeze-dried in a vacuum freeze dryer precooled at −80 °C in advance. If supercritical drying was adopted, the solvent was replaced with ethanol 3 times before drying. The cellulose aerogel samples were prepared for measurement.

As a comparison, cellulose aerogels were prepared with ionic liquid as solvent, showing as follows. The ionic liquid selected in this study was [AMIm]Cl. A total of 20 g [AMIm]Cl was added into the beaker, and 0.8 g cellulose powder was added into the [AMIm]Cl. The mixture was stirred in an oil bath at 90 °C until the cellulose powder was completely dissolved to form a sol. Then, the sol was placed in an oven at 60 °C for 4 h to eliminate tiny bubbles. The preparation of cellulose aerogels in ionic liquid did not require a separate process to form gel. The deionized water was slowly added into the sol for solvent replacement. After the solvent exchange, liquid nitrogen was added to the beaker to quickly freeze the cellulose hydrogel, and it was freeze-dried in a vacuum freeze dryer precooled at −80 °C in advance. The cellulose aerogel samples were prepared for comparison.

### 2.3. Characterization

The morphology of all samples in this study was observed by field emission scanning electron microscope (SU-8010, HITACHI, Tokyo, Japan). The X-ray diffraction (XRD) of the raw cellulose powder and cellulose aerogels prepared under different conditions was performed using an X-ray diffractometer (Bruker AXS D8 X-ray diffractometer, Karlsruhe, Germany) at a 35 kV electric potential, 30 mA voltage, and detector angle of 2θ (5~80 degree). The samples were pretreated under vacuum conditions by a standard degassing station equipped with Micromeritics instruments. Additionally, the nitrogen adsorption–desorption test was carried out by a 4-station full-automatic specific surface area analyzer (American Micromeritics APSP 2460, Norcross, GA, USA) under 77 k liquid nitrogen conditions. The isotherm adsorption–desorption curve was obtained, and the total specific surface area of the materials was achieved through the BET method after the instrument analysis had been completed. Thermal analysis was performed on a thermogravimetric analysis instrument (NETZSCH TG 209 F1 Libra, Selb, Germany), including thermogravimetric analysis (TG) and differential thermal gravity (DTG). All samples were recorded at a range of 50–1000 °C under nitrogen atmosphere with a heating rate of 20 °C·min^−1^. The bulk density (ρ) of cellulose aerogels was calculated by a simple method. The cellulose aerogel was cut into regular cuboids, it was weighed to obtain its mass (m), and the side length was measured to calculate the volume (V). The bulk density of cellulose aerogel was calculated according to the corresponding equation (ρ = m/V).

## 3. Results and Discussion

### 3.1. Preparation Mechanism of Cellulose Aerogel in NaOH/Urea Solution

Figure 1 is the schematic illustration of the preparation of cellulose aerogel preparation in NaOH/urea solution. Firstly, NaOH and urea are dissolved in water to form a mixed solution, and then the cellulose is added into the solution under continuous stirring. NaOH “hydrate” is more easily attracted by the cellulose chain through the formation of a new hydrogen bond network, which is relatively stable at low temperatures, while urea embeds NaOH and cellulose connected by a hydrogen bond in the form of a shell to generate an inclusion compound with a sheath structure, which leads to the dissolution of cellulose and the formation of sol [25]. When the sol is heated at the proper temperature, the hydrogen bond between cellulose molecules becomes a more stable connection mode, and the cellulose molecules are gathered in the sol to form the gel. The deionized water is used for solvent replacement to wash NaOH and urea in the wet gel, and the wet gel is quickly frozen with liquid nitrogen. The growth of ice crystals will destroy the cellulose network and squeeze the cellulose network between the ice crystals for aggregation [26]. Finally, the cellulose aerogel with a lamellar network structure can be obtained through freeze-drying.

### 3.2. Cellulose Aerogels Prepared with NaOH/Urea Solution as Solvent

In this preparation procedure with NaOH/urea solution as solvent, we investigate the effects of cellulose concentration and aging time on the micromorphology of cellulose aerogels. Figure 2 shows SEM images of cellulose aerogels prepared with different cellulose concentrations (1 wt%, 2 wt%, 3 wt%, 4 wt%, respectively). It is noted that the cellulose concentration plays an important role in the micromorphology of cellulose aerogels. When the cellulose concentration is 1 wt%, the pore size of the cellulose aerogel is very large, reaching more than 20 μm, and a bulk density of 0.1062 g·cm^−3^ can be achieved with thicker skeletons. When the cellulose concentration increases to 2 wt%, the skeleton thickness and the pore size are significantly reduced, and the pore size decreases to about 10 μm. The bulk density also decreases, becoming 0.0905 g·cm^−3^. When the cellulose concentration reaches 3 wt%, the skeleton thickness and the pore size further decrease, the pore size dwindles to below 5 μm, and the bulk density falls to 0.0736 g·cm^−3^. With the further increase of the cellulose concentration to 4 wt%, the micromorphology of the cellulose aerogel almost does not change anymore, while the bulk density decreases slightly to 0.0675 g·cm^−3^. As a result, with the increase of the cellulose concentration, the skeleton and pore size of cellulose aerogels will decrease. The cellulose concentration can determine the formation of the hydrogel skeleton network of cellulose aerogels. A higher cellulose concentration can form a dense gel skeleton network, which can withstand the growth of ice crystals during freeze-drying and results in the formation of a smaller pore structure and finer skeleton.

Figure 3 shows the SEM images of cellulose aerogels prepared after different aging times. It is observed that the micromorphology of these samples is similar with different aging times. The pore structures are all composed of sheet skeletons, and the pore diameter is 2~3 μm, which indicates that the aging time has no obvious effect on the pore structure of cellulose aerogels. The bulk density of aerogels prepared after aging for 4, 12, 24 and 48 h is 0.0713, 0.0705, 0.0675 and 0.0677 g·cm^−3^, respectively. This implies that the bulk density of cellulose aerogels decreases slightly with the increase of aging time, while the difference is not obvious.

### 3.3. Comparison of Micromorphology and Pore Structure of Cellulose Aerogels

Ionic liquid is a very common solvent for dissolving cellulose, and supercritical drying is often used as a drying method for preparing cellulose aerogels. In this work, we have used a NaOH/urea solution as the solvent and freeze-drying as the drying method (NaOH/urea solution & freeze-drying) to facilely prepare cellulose aerogels. For comparison, the two cellulose aerogel samples were also prepared by using ionic liquid as the solvent and freeze-drying as the drying method (ionic liquid & freeze-drying), and NaOH/urea solution as the solvent and supercritical drying as the drying method (NaOH/urea solution & supercritical drying), respectively.

Figure 4 shows the TG/DTG curves of raw cellulose powder and cellulose aerogels prepared under different conditions when heated in N_2_ atmosphere. From the TG curve, it can be seen that the weight-loss rates of cellulose aerogels prepared under different conditions after complete decomposition are smaller than that of raw cellulose powder. After complete decomposition, the residual weight of raw cellulose powder is about 5%, the residual weight of freeze-dried cellulose aerogel becomes about 15%, while the residual weight of supercritical dried cellulose aerogel reaches about 30%. It can be seen from the DTG curve that the decomposition temperature of raw cellulose powder is the highest, and there is a weight loss peak at about 360 °C, indicating a high decomposition temperature. The decomposition temperature of cellulose aerogels prepared by NaOH/urea solution & freeze-drying is slightly lower, at about 330 °C. The decomposition temperature of cellulose aerogels prepared by ionic liquid & freeze-drying is only 250 °C. In contrast, the cellulose aerogels prepared by NaOH/urea solution & supercritical drying have two weight-loss peaks at 120 and 250 °C, respectively, which can be attributed to the volatilization of adsorbed water and decomposition of cellulose. In combination with XRD patterns, the decomposition temperature is mainly related to the crystallinity of cellulose. A high crystallinity of cellulose can obtain a higher decomposition temperature for cellulose.

Figure 5 shows the X-ray diffraction patterns of raw cellulose powder and cellulose aerogels prepared under different conditions. Cellulose has four different crystalline structures, which can be distinguished by X-ray diffraction. The positions of X-ray diffraction characteristic peaks of cellulose with different crystalline structures are given in Table 1. The X-ray diffraction pattern of raw cellulose powder shows that there are three obvious diffraction peaks at 14.8°, 16.3° and 22.6°, which indicates that it has a Cellulose Ⅰ structure. The X-ray diffraction peaks of cellulose aerogels prepared with NaOH/urea solution as the solvent are 12.1°, 19.8° and 22.0°, which indicates that as-prepared cellulose aerogels belong to a Cellulose Ⅱ structure. The cellulose molecules have enough time to arrange themselves, since the gel process is relatively slow, which causes the regenerated cellulose to have a relatively high crystallinity [27]. The other peaks in the X-ray diffraction patterns of these two samples belong to a small amount of residual NaOH. In contrast, cellulose aerogels prepared in ionic liquid have no obvious diffraction peaks, indicating that there is almost no crystal region between cellulose molecules. As mentioned above, in the ionic liquid system, the cellulose network structure is formed through non-solvent induced phase separation, and the rapid phase separation process does not allow enough time for cellulose to arrange the crystal region [28]. The diffraction peak of the (020) crystal plane is selected to compare the crystallinity of raw cellulose powder and cellulose aerogels prepared under different conditions through the Scherrer equation. One can find that the raw cellulose powder obviously has a larger crystal area, which corresponds to the higher decomposition temperature on the TG curve.

SEM images of raw cellulose powder and cellulose aerogels obtained by different conditions are shown in Figure 6. One can obviously see that the cellulose powder has a fibrous structure with a length of about 50 μm and a diameter of about 10 μm. The cellulose aerogel prepared by ionic liquids & freeze-drying has a fibrous network structure, and most of the pore sizes are about 200 nm. Some mesopores and small-size macropores below 100 nm can also be observed. This reveals that the cellulose network structure in ionic liquid is strong enough to not be damaged by ice crystals during freezing [30]. The cellulose aerogel prepared by NaOH/urea solutions & supercritical drying also possesses a 3D fibrous network structure. The skeletons are thicker than those of cellulose aerogels prepared in ionic liquids, and the overall pore size is smaller, at about 50 nm. In terms of structure, the cellulose aerogel prepared by NaOH/urea solution & freeze-drying is quite different from the former two aerogels, being composed of schistose skeletons with a pore diameter of 2 μm. This is because the growth of ice crystals destroys the original cellulose network structure and squeezes the cellulose network for aggregation.

Figure 7 shows the N_2_ adsorption–desorption isotherms and pore-size distribution of cellulose aerogels prepared under different conditions. According to the IUPAC classification, all samples exhibit the Type Ⅳ isotherm with an obvious hysteresis loop, which indicates that mesopores exist in cellulose aerogels. Additionally, the adsorption capacity increases gently in the low pressure region, indicating that there are almost no micropores in the samples. Combined with the SEM image, one can clearly see the morphology and pore diameter of the three-dimensional network of the aerogel samples. The specific surface area of cellulose aerogels prepared by ionic liquid & freeze-drying is 42.14 m^2^·g^−1^. The pore size of this cellulose aerogel is mainly 100~200 nm, and there are also some mesopores with a pore size of about 30 nm. The overall pore size distribution of cellulose aerogel prepared by NaOH/urea solution & supercritical drying is relatively concentrated, at about 30 nm, which leads to a relatively high specific surface area of 96.58 m^2^·g^−1^. The cellulose aerogel prepared by NaOH/urea solution & freeze-drying is dominated by a macroporous structure of about 2 μm, while there are a few mesoporous structures. The mesoporous structure contributes more to the surface area, promoting its specific surface area up to 27 m^2^·g^−1^.

The pore-size data obtained by BET test is below 100 nm, and the test results of micropores and mesopores are generally accurate. Cellulose aerogels have a large number of macroporous structures, especially cellulose aerogels prepared by NaOH/urea solution & freeze-drying, as shown in Figure 8. It is observed that the macropore sizes of cellulose aerogels prepared by NaOH/urea solution & freeze-drying mainly cover 1~3 μm and that the peak pore size appears at 1.62 μm. Since cellulose aerogel will be compressed to a certain extent during pressurization, the measured pore size is slightly smaller than the actual value. This test can only conduct a rough analysis of the pore structure of cellulose aerogels.

## 4. Conclusions

In summary, ionic liquid and NaOH/urea solution were used as the solvents of cellulose, and several cellulose aerogels were prepared by sol-gel method combined with freeze-drying and supercritical drying, respectively. With NaOH/urea solution as the solvent, the cellulose concentration has a great influence on the structure and density of cellulose aerogels. With an increase in the cellulose concentration, the skeletons and the pore size of cellulose aerogels decrease, as does the density of cellulose aerogels. Using ionic liquid as the solvent or supercritical drying as the drying method will reduce the pore size of cellulose aerogels. The raw cellulose powder has a Cellulose I structure with a high crystallinity and high decomposition temperature, while the cellulose of each cellulose aerogel has a Cellulose II structure with a weak crystallinity and low decomposition temperature.

## Figures and Tables

**Figure 1 nanomaterials-13-00613-f001:**
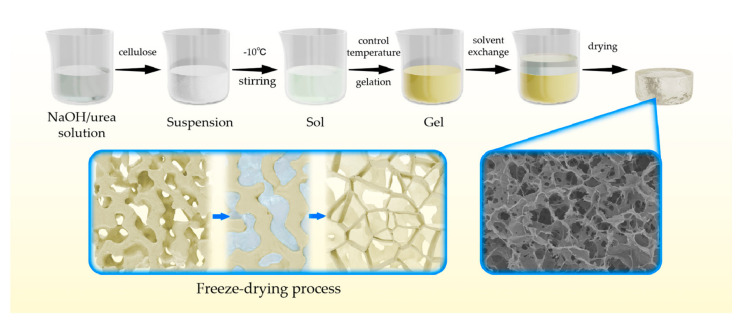
Schematic illustration of cellulose aerogel preparation in NaOH/urea solution.

**Figure 2 nanomaterials-13-00613-f002:**
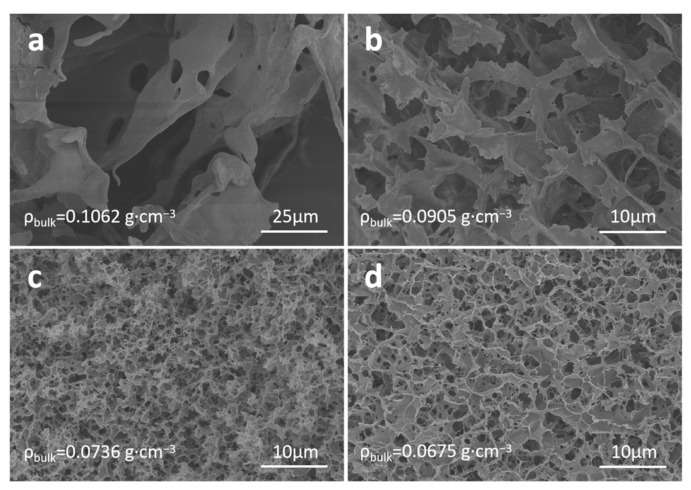
SEM images of cellulose aerogels prepared in NaOH/urea solution with different cellulose concentrations: (**a**) 1 wt%, (**b**) 2 wt%, (**c**) 3 wt%, and (**d**) 4 wt%.

**Figure 3 nanomaterials-13-00613-f003:**
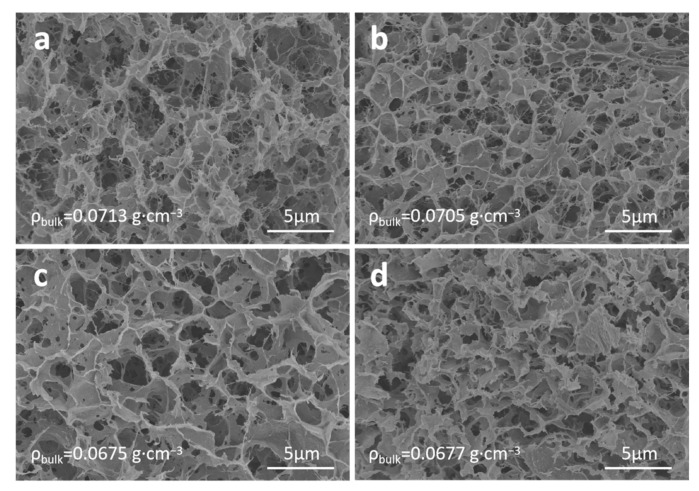
SEM images of cellulose aerogels prepared in NaOH/urea solution after different aging times: (**a**) 4 h, (**b**) 12 h, (**c**) 24 h, and (**d**) 48 h.

**Figure 4 nanomaterials-13-00613-f004:**
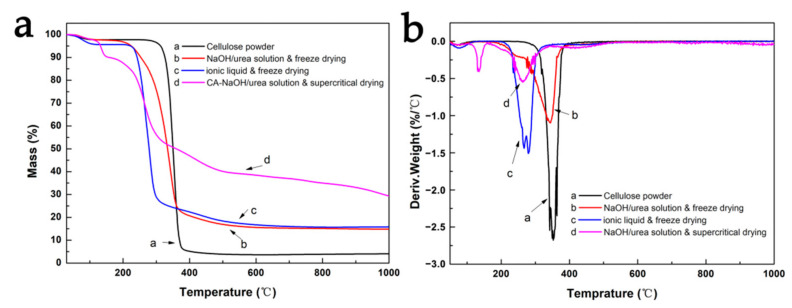
(**a**) TG curve and (**b**) DTG curve of raw cellulose powder and cellulose aerogels prepared under different conditions.

**Figure 5 nanomaterials-13-00613-f005:**
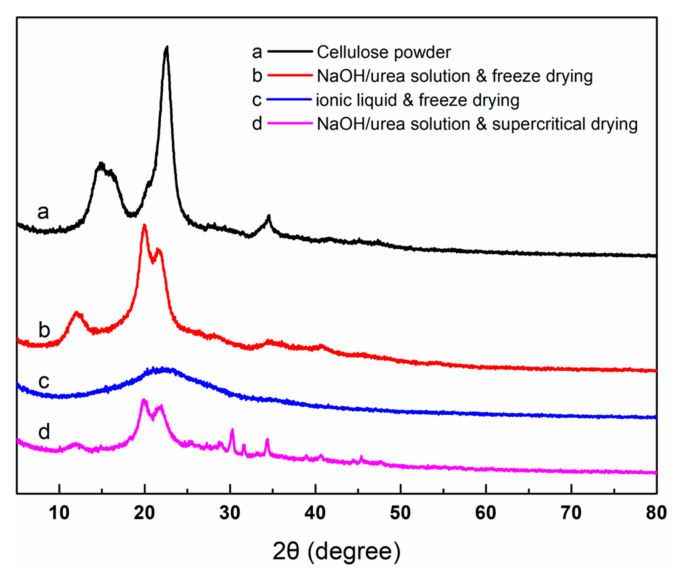
X-ray diffraction patterns of raw cellulose powder and cellulose aerogels prepared under different conditions.

**Figure 6 nanomaterials-13-00613-f006:**
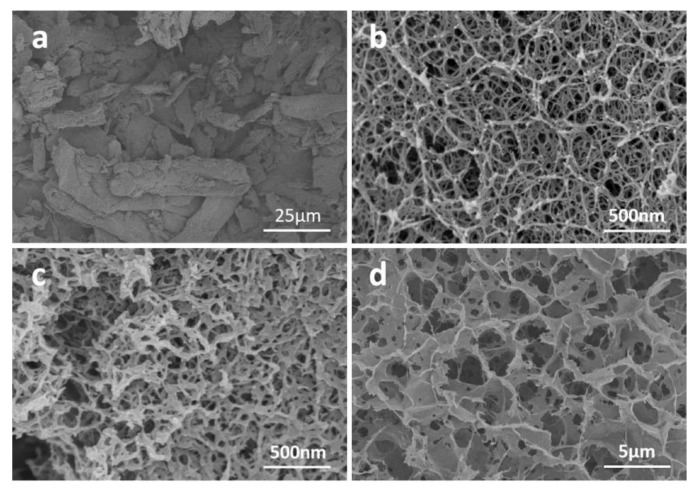
SEM images of (**a**) raw cellulose powder and cellulose aerogels prepared under different conditions: (**b**) ionic liquids & freeze-drying, (**c**) NaOH/urea solutions & supercritical drying, and (**d**) NaOH/urea solutions & freeze-drying.

**Figure 7 nanomaterials-13-00613-f007:**
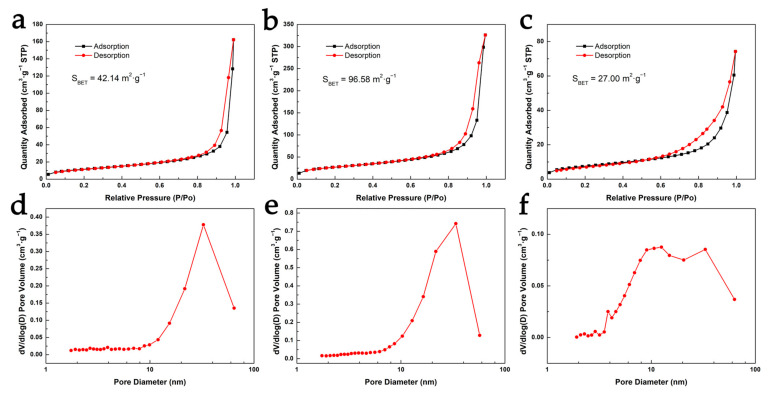
(**a**–**c**) N_2_ adsorption–desorption isotherms and (**d**–**f**) BJH pore-size distribution of cellulose aerogels prepared under different conditions: (**a**,**d**) ionic liquid & freeze-drying, (**b**,**e**) NaOH/urea solution & supercritical drying, and (**c**,**f**) NaOH/urea solution & freeze-drying. The BET result of cellulose aerogels is respectively inserted in the figures.

**Figure 8 nanomaterials-13-00613-f008:**
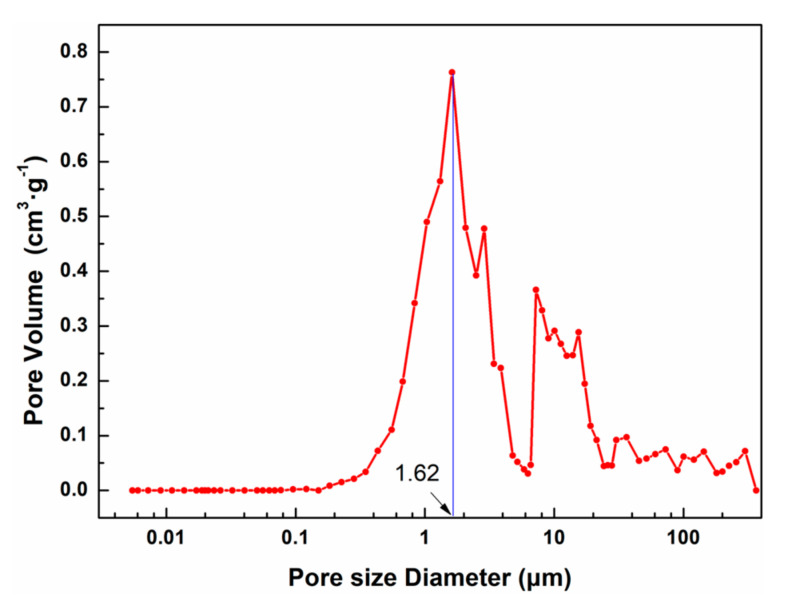
Macropore size distribution (as shown in the red line) of cellulose aerogels prepared by using NaOH/urea solution as solvent via freeze-drying by mercury intrusion method. The blue line shows the pore size corresponding to the peak.

**Table 1 nanomaterials-13-00613-t001:** Peak positions in X-ray diffraction of cellulose polymorphs [29].

Polymorph	Diffraction Angle 2θ, Degree		
	1$\overset{–}{1}$0	110	020
Cellulose Ⅰ	14.8	16.3	22.6
Cellulose Ⅱ	12.1	19.8	22.0
Cellulose Ⅲ_Ⅰ_	11.7	20.7	20.7
Cellulose Ⅲ_Ⅱ_	12.1	20.6	20.6
Cellulose Ⅳ_Ⅰ_	15.6	15.6	22.2
Cellulose Ⅳ_Ⅱ_	15.6	15.6	22.5

## Data Availability

All data that support this work are included in this manuscript.

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
