# Peer review of "Facile Preparation of Cellulose Aerogels with Controllable Pore Structure"

_nanomaterials, 2023, doi:10.3390/nano13030613_

Round 1

Reviewer 1 Report

Comments to the Author

The authors reported the manuscript entitled “Facile Preparation of Cellulose Aerogels with Controllable Pore Structure” The manuscript is well organized, however, there are still some points that need to be addressed. I recommend acceptance of this paper after minor revision.

 Comments:

1.      The author should explain the main objectives of the manuscript and how it differs from the previously reported work. What is the main novelty of the present work?

2.      The author should summarize the type of aerogels prepared, its relevance with cellulose aerogels, and the superiority of cellulose aerogels over the other type.

3.      What are the renewable and biocompatible advantages of cellulose aerogels?

4.      What is the effect of NaOH/urea concentration and aging time on the crystallinity, lattice parameter, and volume using XRD and SEM results analysis? Using the Scherrer equation the author should describe the relevance of the parameters. The author gives proper credit to (Ex. https://doi.org/10.1002/er.7501, https://doi.org/10.1016/j.cjph.2017.12.024, and https://doi.org/10.1016/j.surfcoat.2015.09.021) evaluate these parameters

5.      The author should define all the used abbreviations at the end of the manuscript in accordance with the journal format.

6.      There are some grammatical errors that should be removed, the author should check the manuscript very carefully. 

Reviewer 2 Report

In this article, the authors have prepared cellulose aerogel by using NaOH/urea solution as solvent, rising temperature to control gelation and drying of wet gel sequentially. Authors have characterized the cellulose aerogel with XRD, nitrogen adsorption desorption and, thermogravimetric analysis and bulk density. The authors concluded that aerogel prepared with NaOH/Urea has great influence on the structure and density and while using ionic liquid the pore size reduced for cellulose aerogel. In the current reviewer’s opinion, the manuscript may be considered for a major revision.

Comments to the Authors 

1.     In the introduction authors may give more details cellulose aerogel using ionic liquids.

2.     In preparation method authors may include exact amount of water and NaOH taken for the experiment for the better understanding of the readers.

3.     Authors have to clearly mention exact quantity used for the dissolution of cellulose aerogel using [AMIm]Cl.

4.     Authors may explain why the crystallinity of the aerogel decreased while using ionic liquid as solvent with references.

5.     Include the reason why the pore size decreased while using ionic liquid with suitable references.

6.     Explanation is lacking why ionic liquid is better solvent to prepare the aerogel in compared to the NaOH/Urea solution in discussion.

Reviewer 3 Report

The manuscript reports the fabrication of cellulose aerogel with controllable pore structure based on the use of NaOH/urea solution. The effects of cellulose concentration and aging time on the pore structure of fabricated aerogel were investigated. The fabricated cellulose aerogel shows stronger dependence on cellulose concentrations used in NaOH/urea solution. No significant influence of aging time was found.

The reported results of the parametric study appear to be reasonably presented. However, the mechanism or detailed explanation of the cellulose/NaOH/urea system on the specific formation of porous structure seems lacking. The motivation and originality of the manuscript are also unclear, i.e. the manuscript seems to focus on process optimization or parametric study. For examples,

1)      Figure 5 presents the formation of different crystalline structures by fabrication methods. The authors obtained a cellulose II structure with NaOH/urea and an amorphous structure when using an ionic liquid. However, no detailed explanation of getting different structures was given. The conclusion section summarizes the results, but no discussion on the mechanism was found in the manuscript.

2)      The authors found the dominant formation of the macropore sizes of cellulose aerogels by the used NaOH/urea approach. The authors need to add plausible reasons for the dominant macroporous structure. Is the cellulose aerogel with macroporous structure better for the applications?

Round 2

Reviewer 2 Report

Manuscript may be accepted.

Reviewer 3 Report

The authors explained that the gelation process promotes crystallinity. It is recommended to add supporting references on the crystallinity change driven by the arrangement time of the crystal region. Additionally, the references to show the destruction of the original cellulose network by the ice crystals can be necessary in the manuscript.
